# Deep mutational scans of XBB.1.5 and BQ.1.1 reveal ongoing epistatic drift during SARS-CoV-2 evolution

**Ashley L. Taylor, Tyler N. Starr** *

Department of Biochemistry, University of Utah School of Medicine, Salt Lake City, Utah, United States of America

* tyler.starr@biochem.utah.edu

## Abstract

Substitutions that fix between SARS-CoV-2 variants can transform the mutational landscape of future evolution via epistasis. For example, large epistatic shifts in mutational effects caused by N501Y underlied the original emergence of Omicron, but whether such epistatic saltations continue to define ongoing SARS-CoV-2 evolution remains unclear. We conducted deep mutational scans to measure the impacts of all single amino acid mutations and single-codon deletions in the spike receptor-binding domain (RBD) on ACE2-binding affinity and protein expression in the recent Omicron BQ.1.1 and XBB.1.5 variants, and we compared mutational patterns to earlier viral strains that we have previously profiled. As with previous deep mutational scans, we find many mutations that are tolerated or even enhance binding to ACE2 receptor. The tolerance of sites to single-codon deletion largely conforms with tolerance to amino acid mutation. Though deletions in the RBD have not yet been seen in dominant lineages, we observe tolerated deletions including at positions that exhibit indel variation across broader sarbecovirus evolution and in emerging SARS-CoV-2 variants of interest, most notably the well-tolerated Δ483 deletion in BA.2.86. The substitutions that distinguish recent viral variants have not induced as dramatic of epistatic perturbations as N501Y, but we identify ongoing epistatic drift in SARS-CoV-2 variants, including interaction between R493Q reversions and mutations at positions 453, 455, and 456, including F456L that defines the XBB.1.5-derived EG.5 lineage. Our results highlight ongoing drift in the effects of mutations due to epistasis, which may continue to direct SARS-CoV-2 evolution into new regions of sequence space.

## Author summary

SARS-CoV-2 variants evolve in part via mutations in the spike receptor-binding domain (RBD) that impact the ability of this domain to evade binding by neutralizing antibodies while maintaining high-affinity binding to ACE2 receptor. To aid in ongoing viral forecasting and surveillance, we conducted high-throughput measurements of the impacts of all possible amino acid mutations or single-codon deletions on ACE2 binding in the

**Data Availability Statement:** Site saturation mutagenesis libraries and respective isogenic parental plasmid stocks are available from Addgene (Addgene Cat# 208085, 208086, 1000000231, and 1000000232; all libraries

available at: https://www.addgene.org/pooled-library/bloom-sars-cov-2-rbd-ssm/). Raw sequencing data are on the NCBI SRA under BioProject PRJNA770094, BioSamples SAMN37185589 (PacBio sequencing) and SAMN37185929 (Illumina barcode sequencing for ACE2 binding and expression experiments). All code and data at various stages of processing is available at https://github.com/tstarrlab/SARS-CoV-2-RBD_DMS_Omicron-XBB-BQ/tree/main. Outlines of the analytical pipelines and links to descriptive notebooks for each analytical step are available at https://github.com/tstarrlab/SARS-CoV-2-RBD_DMS_Omicron-XBB-BQ/blob/main/results/summary/summary.md. Final mutant deep mutational scanning measurements are available in Supplemental Data 1, and interactive visualizations of key data are available at: https://tstarrlab.github.io/SARS-CoV-2-RBD_DMS_Omicron-XBB-BQ/.

**Funding:** This work was supported in part by grants from the National Institute of Allergy and Infectious Diseases, National Institutes of Health, Department of Health and Human Services (K99AI166250 and Contract No. 75N93021C00015 to T.N.S.) and a Dale F. Frey Breakthrough Scientist Award from the Damon Runyon Cancer Research Foundation (to T.N.S.). The funders had no role in study design, data collection and analysis, decision to publish, or preparation of the manuscript.

**Competing interests:** I have read the journal's policy and the authors of this manuscript have the following competing interests: T.N.S. consults for Apriori Bio, Metaphore Biotechnologies, and Vir Biotechnology on deep mutational scanning. The lab of T.N.S. has received sponsored research agreements unrelated to the present work from Vir Biotechnology and Aerium Therapeutics, Inc. T.N.S. may receive a share of intellectual property revenue as inventor on a Fred Hutchinson Cancer Center-optioned patent related to stabilization of SARS-CoV-2 RBDs.

newly evolved Omicron BQ.1.1 and XBB.1.5 variant backgrounds. We find that mutations and deletions are well-tolerated in these domains, consistent with the ongoing evolutionary potential of Omicron sub-variants. We show that the impacts of mutations on ACE2 binding continue to change over time due to the phenomenon of epistasis, though these shifts in mutational effect are less pronounced than epistatic shifts described in earlier variants of concern. Nonetheless, we show that this epistasis continues to enable SARS-CoV-2's exploration of new mutational combinations as it evolves into new regions of sequence space, highlighting the ongoing evolutionary potential this virus will continue to exhibit.

## Introduction

Global evolution of SARS-CoV-2 features repeated turnover of major circulating variants containing amino acid substitutions that impact molecular phenotypes [1]. Though substitutions across the viral genome influence variant fitness, the spike receptor-binding domain (RBD) has shown a particularly high rate of sequence evolution due to its role in binding ACE2 receptor to enable cell entry [2,3] and its dominant recognition by neutralizing antibodies [4,5]. RBD sequence evolution is therefore driven in large part by pressure to escape population antibody-based immunity while maintaining or restoring high-affinity binding to ACE2 receptor [6].

Deep mutational scanning has proven a useful method to experimentally measure the impacts of all possible single mutations in viral spike glycoproteins [7]. We and others have conducted deep mutational scans to prospectively query the impacts of SARS-CoV-2 mutations on ACE2-binding, cell entry, protein folding, and antibody escape [6,8–16], enabling real-time evaluation of the phenotypic impacts of mutations seen during global viral surveillance. These studies have revealed a general tolerance to mutations in the SARS-CoV-2 RBD, which presumably underlies the ability of this domain to continually evolve to escape antibody recognition while maintaining ACE2 binding.

However, the phenotypic effects of mutations can change across genetic backgrounds due to the phenomenon of epistasis (generally defined as non-additivity of mutational effects) [17]. For SARS-CoV-2, epistasis has caused measurable shifts in mutations' effects across variant backgrounds on ACE2 binding [6,18–22] and antibody recognition [18,23,24]. For example, the RBD mutation N501Y, which has occurred convergently in many SARS-CoV-2 variants including Alpha, Beta, and Omicron, alters the impacts of mutations at many other RBD sites, including those that underlied Omicron's emergence [6,19,20]. Understanding these patterns of epistatic shifts in mutational effects over SARS-CoV-2 evolution is therefore key for ongoing modeling and forecasting of SARS-CoV-2 evolution.

Since the initial appearance of the Omicron BA.1 variant in November 2021 [25], a series of Omicron sub-variants have proliferated including BA.2, BA.4, BA.5, and more recently BQ.1.1 and XBB.1.5 (Fig 1A). Omicron BQ.1.1 and XBB.1.5 variants evolved in late 2022 within a milieu of Omicron sub-variants that were convergently sampling RBD mutations to maximize antibody escape [26–28]. XBB.1.5 rose to dominate case counts in many regions for the first half of 2023 [29] due in part to its ability to balance strength of ACE2 binding with immune escape via the F486P mutation (which required a two-nucleotide codon change from the ancestral F486 residue through an F486S intermediate). Following its growth in case counts, XBB.1.5 has left a trail of derivative lineages in its wake. Although we currently have tools to predict the impacts of ongoing Omicron sequence evolution on antibody escape phenotypes [28,30], we lack an updated understanding of how these mutations are constrained or facilitated by their secondary effects on ACE2-binding affinity in these newest viral sequence backgrounds.

Here, we use deep mutational scanning to measure the impacts of all amino acid changes and single-codon deletions in the BQ.1.1 and XBB.1.5 RBDs on ACE2-binding affinity and RBD folding efficiency. We compare our measurements of mutational effects in these new variant backgrounds to those measured previously in a broader set of SARS-CoV-2 and sarbecovirus RBDs, revealing the ongoing influence of epistasis in SARS-CoV-2 evolution. Our results suggest that the large-scale shifts induced by N501Y remain an anomaly in the scale of epistatic perturbation induced by a single substitution, but we identify ongoing epistatic drift [31] in the impacts of amino acid changes on ACE2 receptor binding that may continue to project SARS-CoV-2 evolution into new regions of sequence space.

## Results

### Deep mutational scanning of the Omicron BQ.1.1 and XBB.1.5 RBDs

We conducted deep mutational scanning experiments to measure the impacts of all amino acid changes and single-codon deletions in the Omicron BQ.1.1 and XBB.1.5 RBDs on ACE2-binding affinity and RBD expression. We constructed duplicate site-saturation mutagenesis libraries in each RBD background (S1 Fig), tagged each RBD variant with a unique molecular barcode, and cloned our libraries into a yeast-surface display platform [8,32]. We experimentally determined the impact of each RBD mutation on ACE2-binding affinity by incubating yeast-displayed RBD libraries across a concentration gradient of monomeric human ACE2, partitioning each sample on the basis of ACE2-binding via fluorescence-activated cell sorting (FACS), and deep sequencing molecular barcodes from each FACS partition to track the titration of each library variant and determine its dissociation constant ($K_D$) (S2 Fig and S1 Data) [8,33]. An analogous FACS-seq process was used to measure the impact of each RBD mutation on yeast-surface display levels, a proxy for folding efficiency, based on fluorescence detection of a C-terminal epitope tag (S3 Fig and S1 Data).

Heatmaps illustrating the effect of each mutation on ACE2-binding affinity, alongside previously measured datasets in Omicron BA.2 and the ancestral Wuhan-Hu-1 strain, are shown in Fig 1B (and as an interactive figure at https://tstarrlab.github.io/SARS-CoV-2-RBD_DMS_Omicron-XBB-BQ/RBD-heatmaps/). As in prior RBD backgrounds, we find that many sites in the BQ.1.1 and XBB.1.5 RBDs are highly tolerant to mutation. This tolerance to mutation is not a simple consequence of the insignificance of many sites for RBD structure and function, as residues that are in direct structural contact with ACE2 (blue indicator squares, Fig 1B, bottom) or that are dominant sites for escape from monoclonal antibodies (orange indicator squares, Fig 1B, bottom) exhibit considerable mutational plasticity.

Furthermore, as in prior RBD backgrounds, we find many mutations in the BQ.1.1 and XBB.1.5 RBDs that enhance ACE2-binding affinity. There is no evidence that greatly enhanced ACE2-binding affinity per se is beneficial for viral fitness, as successful SARS-CoV-2 variants typically have affinity in a relatively narrow range [34] despite availability of affinity-enhancing mutations that could drive tighter binding if it were preferred. However, mutations that enhance ACE2-binding affinity can have a key role in SARS-CoV-2 variant evolution by offsetting mild ACE2-binding deficits of antibody-escape mutations to restore ACE2 binding [6,18]. Therefore, the continued presence of affinity-enhancing mutations in the BQ.1.1 and XBB.1.5 landscapes will likely facilitate ongoing antigenic evolution via compensatory epistasis.

### Tolerance to single-codon deletions

In contrast to our deep mutational scans on prior RBD variants, our current libraries include single-codon deletions, whose effects are well-correlated between BQ.1.1 and XBB.1.5 backgrounds (S2D Fig). Consistent with the dramatic biochemical consequences of residue

A

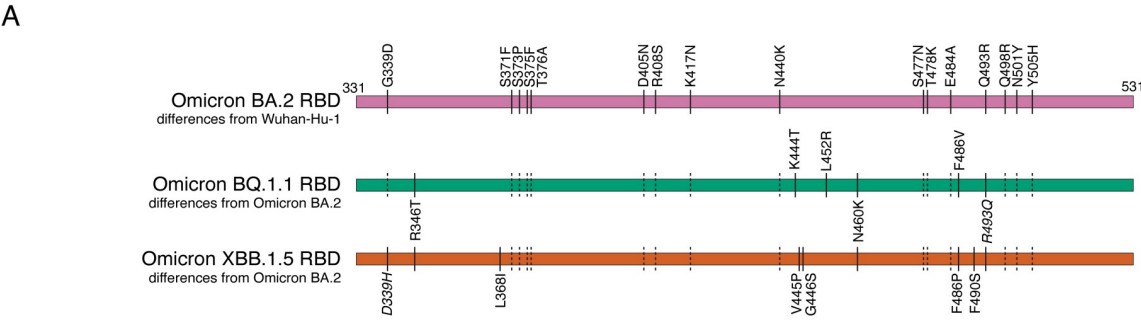

B

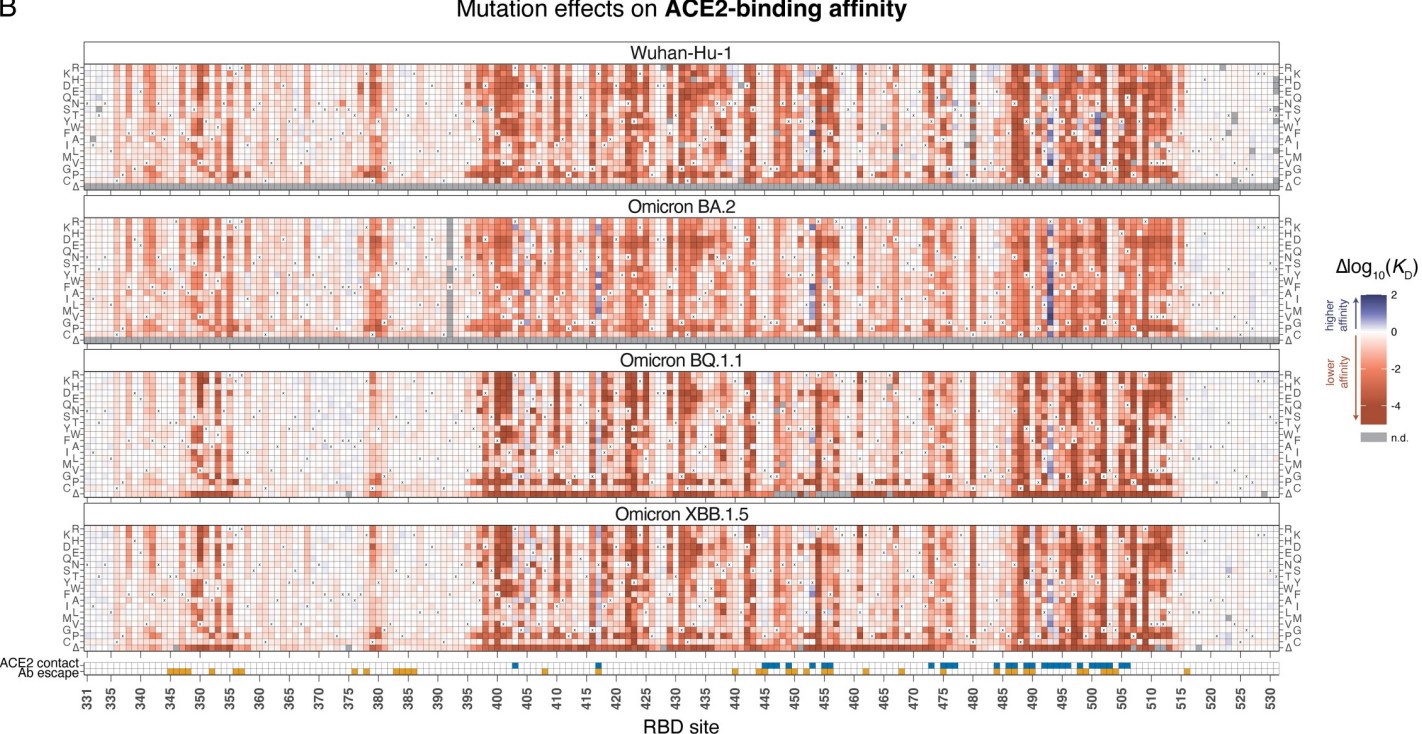

**Fig 1. Effects of mutations in Omicron BQ.1.1 and XBB.1.5 receptor-binding domains on ACE2 binding.** (**A**) Diagram of the RBD substitutions that distinguish BA.2 from Wuhan-Hu-1 (top), and BQ.1.1 and XBB.1.5 from BA.2 (bottom). Dashed black lines illustrate the propagation of substitutions from Wuhan-Hu-1 to BA.2 in BQ.1.1 and XBB.1.5. Italicized mutations in BQ.1.1 and XBB.1.5 show secondarily mutated (D339H) or reverted (R493Q) substitutions that originally changed from Wuhan-Hu-1 to BA.2. Wuhan-Hu-1 reference spike numbering is used throughout the manuscript. (**B**) Heatmaps illustrate the impacts of all mutations and single-codon deletions in the BQ.1.1 and XBB.1.5 RBDs on ACE2-binding affinity as determined from FACS-seq experiments with yeast-displayed RBD mutant libraries. Prior measurements in the Wuhan-Hu-1 and Omicron BA.2 background from [6,18] included for reference. ACE2 contact residues defined as RBD residues with non-hydrogen atoms <5Å from ACE2 in the Wuhan-Hu-1 (PDB 6M0J), BQ.1.1 (PDB 8IF2), and/or XBB.1 (PDB 8IOV) structures. Antibody escape residues defined as those with >0.1 average antibody escape from aggregated deep mutational scanning data [30]. See S1–S3 Figs for additional experimental details. Individual measurements are reported in S1 Data, and an interactive version of these heatmaps is available at https://tstarrlab.github.io/SARS-CoV-2-RBD_DMS_Omicron-XBB-BQ/RBD-heatmaps/.

deletions, we found no sites where the single-codon deletion is substantially better tolerated than the average amino acid mutation at that site (Fig 2A). However, for some sites the deletion is better tolerated than the worst possible amino acid change (Fig 2B), indicating that deletion of a residue is not always the most dramatic biochemical perturbation possible at a site.

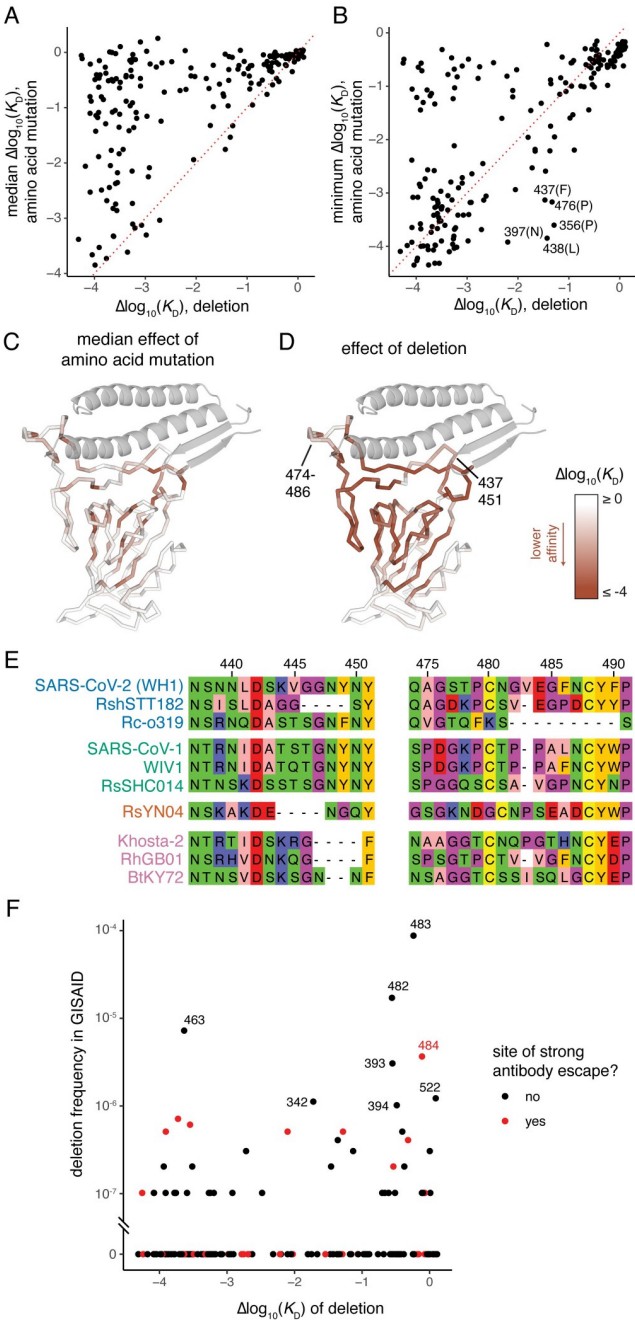

**Fig 2. Effects on ACE2-binding and evolution of RBD amino acid deletions.** (**A, B**) For each RBD site, comparison of the effect on ACE2 binding of single-residue deletion and the median (A) or minimum (B) impact of amino acid mutations at the site as measured in the XBB.1.5 dataset. (**C, D**) Mapping of the effect on ACE2 binding of the median amino acid mutation (C) or single-residue deletion (D) onto the ACE2-bound RBD structure. (**E**) Select alignment of ACE2-utilizing sarbecovirus RBDs illustrating patterns of deletion in loops at the ACE2-binding interface. Alignment numbering according to SARS-CoV-2 Wuhan-Hu-1 reference. Sarbecovirus variants are clustered and colored according to phylogenetic clades of RBD sequence [41]. (**F**) Comparison of the frequency of deletions at each RBD residue in the GISAID database of SARS-CoV-2 genomic sequences with the measured impact on ACE2 binding in the XBB.1.5 background. Sites of strong antibody escape, as defined in Fig 1B, are highlighted in red.

Although RBD residues at the ACE2-interaction surface are remarkably tolerant to amino acid mutation (Fig 1B [bottom row of heatmaps] and 2C), single-codon deletions at the interface are often quite detrimental for ACE2 binding (Fig 2D). However, two peripheral loops in the RBD comprising residues 474–486 and 437–451 are modestly tolerant to deletion. Notably, these loops contain many residues that are frequent targets of neutralizing antibodies (e.g., residues 440, 444, 445, 484, and 486). These loops also exhibit deletions over longer-term evolution of ACE2-utilizing bat SARS-related coronaviruses (Fig 2E). For example, Δ483 is present in SARS-CoV-1 and related bat sarbecoviruses, as well as the SARS-CoV-2-related RshSTT182 isolate from Cambodia and sarbecoviruses from Europe such as RhGB01. A larger deletion of residues Δ482–490 is also present in the ACE2-utilizing Rc-o319 and related sarbecoviruses from Japan, and several independent deletions between 2 and 4 codons in length between residues 444–450 are found among ACE2-utilizing sarbecovirus lineages (Fig 2E). This deeper alignment highlights the potential for deletions in these two loops to fix during viral evolution while maintaining ACE2 receptor usage.

Amino acid deletions have evolved in many SARS-CoV-2 variants in the spike N-terminal domain (NTD) [35,36], but thus far no dominant variant lineages have fixed deletions in the RBD. We queried the >16 million SARS-CoV-2 genomes present on GISAID as of November 11, 2023, eliminated sequences with many deletions indicative of sequencing errors, and identified a handful of RBD single-codon deletions that rise to as high as ~1/10,000 frequency (Fig 2F). Notably, many of the deletions with modest frequencies are well-tolerated for ACE2-binding suggesting that they could have the potential to evolve in the future, for example due to impacts on antibody escape [12]. Although the frequency of many deletions tracks with their measured tolerability for ACE2 binding, several deletions that are very detrimental for ACE2 binding are also present at modest frequencies. Despite our attempts to eliminate genome accessions with poor sequencing quality, these modestly frequent but detrimental mutations could reflect sequencing errors, though they also contain sites of strong antibody escape which indicates a functional explanation for their frequency could also be possible.

Of note, Δ483 is well-tolerated for ACE2 binding and has been seen with modest frequency (Fig 2F), including in carefully tracked wastewater samples that may represent chronic host infections [37,38]. The Δ483 deletion most recently rose to prominence due to its presence in the second-generation BA.2 variant of interest BA.2.86. Although the BA.2.86 variant has not yet risen to a dominant frequency, it has attracted considerable attention due to its sudden appearance in multiple countries and its distant divergence from currently circulating strains going back to a BA.2 common ancestor [39,40]. Though 483 is not directly annotated as a strong site for antibody escape, its proximity to the immunodominant site 484 suggests the Δ483 deletion could have important antigenic consequences.

## Epistatic shifts in Omicron BQ.1.1 and XBB.1.5

We next determined how epistasis alters the effects of mutations on ACE2 binding in BQ.1.1 and XBB.1.5 RBDs compared to prior SARS-CoV-2 variants. We computed an "epistatic shift" metric for each RBD residue [6] that identifies sites where the effects of mutations diverge most dramatically between pairs of variants (Fig 3A; interactive visualization available at https://tstarrlab.github.io/SARS-CoV-2-RBD_DMS_Omicron-XBB-BQ/epistatic-shifts/). This epistatic shift is a probabilistic distance metric (Jensen-Shannon distance) comparing two backgrounds, computed on the vectors of affinities measured for the 20 amino acids possible at a position (21 for BQ.1.1 and XBB.1.5 including deletion mutants). This epistatic shift metric scales from 0 for a site where the measured affinities of each amino acid mutant are identical between two RBD backgrounds to 1 for a site where the distributions are entirely dissimilar.

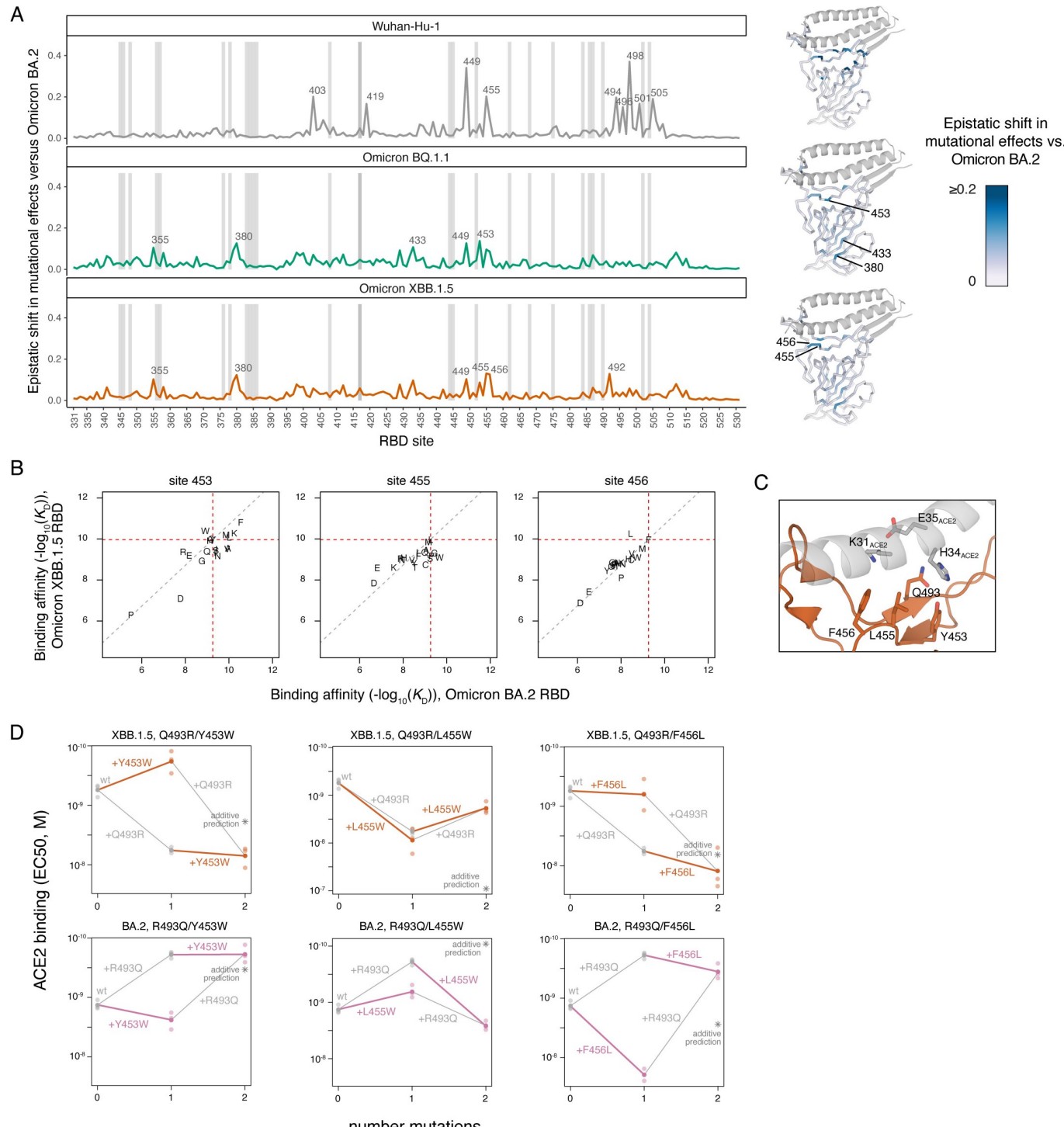

**Fig 3. Epistatic shifts in mutational effects on ACE2 binding.** (**A**) Epistatic shift in the effects of mutations on ACE2 binding at each RBD position as measured in Wuhan-Hu-1, BQ.1.1, or XBB.1.5 compared to those measured in Omicron BA.2. Wuhan-Hu-1 and Omicron BA.2 datasets are included from prior publications [6,18]. Interactive plot is available at https://tstarrlab.github.io/SARS-CoV-2-RBD_DMS_Omicron-XBB-BQ/epistatic-shifts/. Gray lines indicate sites of strong antibody escape, as defined in Fig 1B. Structures at right map the epistatic shift metric to the ACE2-bound RBD. (**B**) Mutation-level plots of epistatic shifts at sites of interest. Each scatterplot shows the measured ACE2-binding affinity of each amino acid in the XBB.1.5 versus BA.2 background (see S4A and S4B Fig for additional sites and backgrounds). Red dashed lines mark the parental RBD affinities, and the gray dashed line indicates the additive (non-epistatic) expectation. (**C**) Zoomed structural view of the ACE2-bound XBB.1 RBD structure (PDB 8IOV), illustrating the structural proximity of residue 493 with residues 453–456. (**D**) Double mutant cycle diagrams illustrating epistatic interactions in the XBB.1.5 (top) or BA.2 (bottom) background (representative

binding curves shown in S4C Fig). Transparent points indicate per-replicate measurements, and solid points and lines connect the averaged binding values for each genotype. Colored lines show the impact of introducing Y453W, L455W, or F456L in the wildtype versus Q493R/R493Q backgrounds, with gray lines indicating the complementary change in effects of 493 mutations. Asterisk indicates expected double-mutant binding affinity assuming additivity of mutational effects.

We previously showed how the substitutions that separate Omicron BA.2 from the ancestral Wuhan-Hu-1 strain induce widespread epistatic shifts across the RBD's ACE2-contact surface (Fig 3A, top). Comparing Omicron BA.2 to BQ.1.1 and XBB.1.5, however, reveals more muted epistatic perturbations (Fig 3A, middle and bottom). Some sites that show modest epistatic shifts in mutational effects map to the beta-sheet structure of the core RBD (e.g., residues 355, 380, 433). At these positions, there are no epistatic changes in the sign of a mutation's effect (i.e., no mutations change from being affinity-enhancing in one background to being affinity-decreasing in another). Instead, these sites show greater sensitization to deleterious mutational effects in the BA.2 background (S4A Fig). This pattern of perturbation, together with the localization of these sites in the buried core of the RBD away from the ACE2-binding interface, is suggestive of nonspecific epistasis [17] caused by the decreased folding stability of the original Omicron BA.1 and BA.2 RBDs that was restored during subsequent Omicron evolution [42,43].

However, we did find a trio of sites (453, 455, and 456) within one of the central strands comprising the ACE2 interface that showed epistatic shifts between BA.2 and BQ.1.1 or XBB.1.5, including prominent sign epistatic changes in the effects of mutations (Figs 3A, 3B and S4B). For example, in our deep mutational scanning datasets the mutations Y453W and F456L decrease ACE2-binding affinity 2.2- and 6.6-fold in the Omicron BA.2 background, respectively, but enhance ACE2-binding affinity 7.1- and 1.9-fold in XBB.1.5 (Fig 3B). In contrast, the mutation L455W enhances ACE2-binding affinity 2.5-fold in BA.2 but decreases affinity 7.4-fold in XBB.1.5 (Fig 3B). Each of these sites exhibits potential for evolution. For example, residues 455 and 456 show substantial amino acid variation across the sarbecovirus alignment, the Y453F mutation occurred independently in multiple outbreaks of SARS-CoV-2 in farms of American mink [44,45], and the mutation L455W has been observed in a chronic human SARS-CoV-2 infection [46]. At the time of writing, XBB.1.5 descendants such as EG.5 and XBB.1.16.6 (each containing F456L) and an EG.5 derivative that combines F456L with L455F (HK.3) have been detected and begun to rise substantially in global frequency consistent with a selective advantage. These observations suggest that the epistatic shifts along this central RBD strand that we measured for ACE2 binding have key roles in shaping ongoing Omicron RBD evolution.

We next wanted to identify the substitutions from Omicron BA.2 to BQ.1.1 and XBB.1.5 that are responsible for these sign epistatic shifts. We noted that residues 453, 455, and 456 are in close proximity to residue 493, which is borne on the complementary strand to that bearing 453–456 at the center of the ACE2-binding interface (Fig 3C). Site 493 has changed across SARS-CoV-2 evolution, with the ancestral glutamine substituting to arginine (substitution Q493R) leading to Omicron BA.1 and BA.2, before reverting (R493Q) convergently in subsequent Omicron subvariants including BQ.1.1 and XBB.1.5 (Fig 1A).

To test whether substitutions at site 493 are responsible for the observed epistatic changes observed in the deep mutational scanning data, we measured ACE2-binding affinities of the Y453W, L455W, and F456L mutations individually and in combination with Q493R or R493Q in the XBB.1.5 and BA.2 backgrounds using isogenic yeast-display titration assays (S4C Fig). In all three cases, we confirmed the hypothesized epistasis (Fig 3D): we found that in both the XBB.1.5 and BA.2 backgrounds, Y453W and F456L mutations had more favorable effects on ACE2-binding when introduced into the Q493 rather than R493 background, while L455W

exhibited reciprocal sign epistasis with site 493 with a favorable effect on ACE2 binding occurring only when introduced together with R493. Additional variation in the strength of magnitude or sign epistasis in the XBB.1.5 and BA.2 backgrounds suggests that these pairwise epistatic interactions are modulated by additional higher-order epistasis with other substitutions between BA.2 and XBB.1.5. These epistatic dissections point to a key role of mutations at site 493 in modulating affinity-enhancing effects of secondary mutations on the 453–456 RBD strand and emphasize the need to monitor variant evolution at these sites with proper epistatic context.

## Gradual versus punctuated epistatic shifts during SARS-CoV-2 evolution

We have now performed deep mutational scans for ACE2-binding affinity in nine SARS-CoV-2 variant backgrounds [6,18], as well as the divergent human-ACE2-utilizing SARS-CoV-1 RBD [47], which shares 74% RBD sequence identity with the ancestral SARS-CoV-2 Wuhan-Hu-1 strain. Comparisons across these datasets presents the opportunity to globally explore the extent and directionality of epistatic drift [31] over SARS-CoV-2 evolution (as opposed to analyses like Fig 3 that compare one specific reference background to another).

We computed an aggregated dissimilarity metric to capture the overall shift in mutational effects on human ACE2 binding for each of the 45 unique RBD pairs. Specifically, we computed epistatic dissimilarity as the root-mean-squared epistatic shift across the 201 RBD sites between a pair of RBD variants. We then used multidimensional scaling to project the matrix of pairwise dissimilarities into a 2-dimensional visualization (Fig 4A). In this visualization, variants that are closer together share similar effects of mutations on ACE2 binding, while variants that are further apart exhibit more extensive epistatic shifts globally across the RBD sequence.

This visualization of epistatic drift captures our prior observation [6] that some early variants such as Delta (RBD mutations L452R+T478K) or Eta (E484K) have similar patterns of mutational effects as the ancestral Wuhan-Hu-1 strain, while other early variants like Alpha (N501Y) and Beta (K417N+E484K+N501Y) show marked epistatic perturbation. We find that distance of SARS-CoV-2 variants in the multidimensional scaling layout of epistatic shifts does not correlate intrinsically with number of sequence changes (Fig 4B), consistent with prior observations that epistatic drift arises from a combination of rare large-effect epistatic modifiers together with consistent small-effect perturbations [31,48,49]. Furthermore, variant jumps in the epistatic landscape can but don't necessarily correspond with large changes in antigenic divergence (Fig 4C), indicating that the important role of epistatic modifiers in SARS-CoV-2

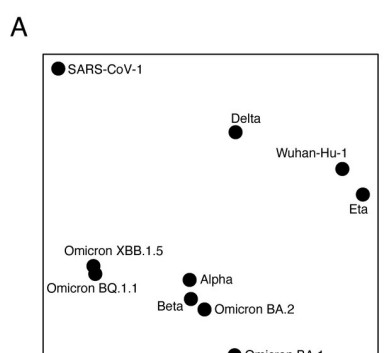
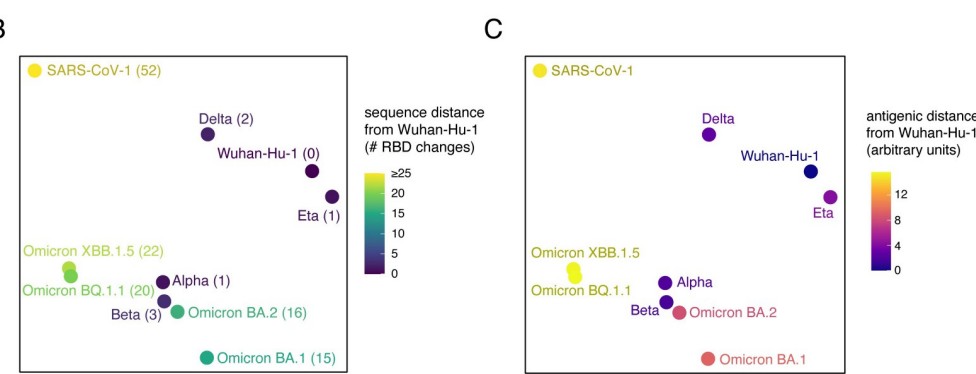

**Fig 4. The epistatic landscape of SARS-CoV-2 evolution.** (**A**) Multidimensional scaling projection of variant backgrounds based on pairwise dissimilarities in per-site epistatic shifts. Dissimilarities between each RBD background were computed as the root-mean-squared epistatic shift across RBD sites (e.g., Fig 3A). (**B**) Layout from (A), with variants colored (and labeled in parentheses) by the number of amino acid differences from the SARS-CoV-2 Wuhan-Hu-1 RBD. (**C**) Layout from (A), with variants colored by antigenic distance from SARS-CoV-2 Wuhan-Hu-1 RBD. Antigenic distances were collated from the literature [27,28], and although displayed in arbitrary units, a one-unit change corresponds approximately to a two-fold loss of neutralization titer.

evolution may not be immune escape per se but rather alterations of the accessibility of secondary immune-escape mutations.

Instead of correlating with overall sequence divergence, we found that the epistatic landscape visualization is dominated by amino acid variation at sites 501 and 498. All of the variants in the bottom-left of the layout fixed the N501Y substitution (which, in the case of Alpha, is the sole RBD difference from Wuhan-Hu-1) together with the ancestral Wuhan-Hu-1 Q498 state or the Omicron Q498R substitution, whereas SARS-CoV-1, which occupies Y498 and T501, gravitates to the opposite corner of the visualization. Extensive epistasis has been described between RBD residues 498 and 501 [6,19–21], for example favoring the R498/Y501 combination, but it has been noted that bulky and/or aromatic substitutions are not tolerated simultaneously at positions 498 and 501 [20,37,38,41]. Individual mutations to aromatic amino acids at residues Q498 or N501 enhance ACE2-binding affinity in the ancestral Wuhan-Hu-1 SARS-CoV-2 strain [8], suggesting that early fixation of Q498Y could have instead driven SARS-CoV-2 closer to SARS-CoV-1 in the epistatic landscape. However, SARS-CoV-2 instead evolved the N501Y substitution, projecting its evolution into new regions of sequence space compared to other sampled sarbecoviruses which have not previously fixed N501Y [41]. Last, even within the subset of Omicron variants that cluster in the N501Y region of the epistatic landscape, we can see that variants continue to diverge in their epistatic niche, suggesting that ongoing sequence divergence of SARS-CoV-2 will continue into the future as it navigates new epistatic combinations of substitutions.

## Discussion

Deep mutational scanning aids in the interpretation and prediction of SARS-CoV-2 evolution. Here, we report updated deep mutational scans in two recent Omicron variant RBD backgrounds, BQ.1.1 and XBB.1.5. Our data illustrate continued tolerance to mutation in the SARS-CoV-2 RBD and provide insight into new mutations that are occurring in the most recent viral variants. Specifically, we show that the Δ483 deletion observed in the novel BA.2.86 variant is well-tolerated for receptor binding despite being near the ACE2-binding interface. We also find that the F456L mutation has fluctuated in its tolerability across variant backgrounds but is particularly well tolerated in XBB.1.5, where it has evolved convergently in multiple variants including those that are currently rising toward dominance. Recent work suggests that F456L itself epistatically alters yet additional mutations' effects [22], highlighting the continued importance of querying mutational effects in new SARS-CoV-2 backgrounds to inform viral monitoring and forecasting.

More broadly, we have now conducted comprehensive RBD mutational scans across many viral backgrounds, enabling insight into the dynamics of epistatic shifts in the RBD over time. Consistent with other studies in molecular evolution, we see that SARS-CoV-2 RBD evolution has sampled a combination of rare, large-effect epistatic saltations with more pervasive small-effect epistatic shifts [31,48,49]. Together, these epistatic interactions act to create cycling windows of mutational accessibility as this virus diverges into new regions of sequence space. Given that such epistasis has allowed some proteins to continue diverging in sequence since their presence in the last universal common ancestor [50], SARS-CoV-2, too, will likely continue to diverge in sequence space for a long time to come.

## Materials & methods

### Mutant libraries

We cloned yeast codon-optimized RBD sequences (amino acids N331 –T531 by Wuhan-Hu-1 reference numbering, the numbering index we use throughout the manuscript) from Omicron

BQ.1.1 and XBB.1.5 into a yeast-surface display plasmid as described [18]. Parental plasmids and associated sequence maps are available from Addgene (Addgene Cat # 208085 and 208086).

Site-saturation mutagenesis libraries spanning all 201 positions in the BQ.1.1 and XBB.1.5 RBDs were produced by Twist Bioscience. We programmed the introduction of precise codon mutations to encode the 19 possible amino acid mutations at each RBD position and a single-codon deletion. To ensure an adequate level of relevant control variants in the library, stop codon mutations were programmed to be introduced at every other position for the first 70 positions, and wildtype codons were specified at every other position for the first 104 positions to ensure sufficient number of wildtype reference variants in each library. Libraries were delivered as dsDNA oligonucleotides with constant flanking sequences. The "mutant RBD fragment" sequence delivered for XBB.1.5 as an example (where uppercase letters denote mutated region) is:

```
tctgcaggctagtggtggaggaggctctggtggaggcggccgcggaggcggagggtcggc
tagccatatgAACATCACCAACTTGTGTCCATTCCATGAAGTTTTCAATGCTACTACTTTC
GCTTCTGTTTACGCTTGGAATAGAAAGAGAATCTCTAACTGCGTTGCTGACTATTCTGTCA
TTTACAATTTTGCTCCATTCTTCGCTTTCAAGTGCTATGGTGTTTCTCCAACTAAGTTGAA
CGATTTGTGTTTCACCAACGTTTACGCCGATTCCTTTGTTATTAGAGGTAACGAAGTCTCC
CAAATTGCTCCAGGTCAAACTGGTAATATTGCCGATTACAATTACAAGTTGCCAGATGATT
TCACCGGTTGTGTTATTGCTTGGAACTCTAACAAGTTGGATTCTAAGCCTTCTGGCAACTA
CAATTACTTGTACAGGTTGTTCCGTAAGTCCAAATTGAAGCCATTCGAAAGAGATATTTCC
ACCGAAATCTATCAAGCTGGTAACAAGCCATGTAATGGTGTTGCTGGTCCTAACTGTTACT
CTCCATTGCAATCTTACGGTTTCAGACCAACTTATGGTGTTGGTCATCAACCATACAGAGT
TGTTGTTTTGTCTTTCGAGTTGTTGCATGCTCCAGCTACTGTTTGTGGTCCAAAGAAATCT
ACTctcgagggggggcggttccgaacaaaagcttatttctgaagaggacttgtaatagagat
ctgataacaacagtgtagatgtaacaaaatcgactttgttcccactgtacttttagctcg
```

A second dsDNA fragment encoding constant flanks and a randomized N16 barcode was produced via PCR off of the parental vector with primer-based sequence additions (primers described in [8,51]). This "barcode fragment" sequence is:

```
cgactttgttcccactgtacttttagctcgtacaaaatacaatatacttttcatttctc
cgtaaacaacatgtttttcccatgtaatatccttttctattttcgttccgttaccaacttt
acacatactttatatagctattcacttctatacactaaaaaactaagacaattttaatttt
gctgcctgccatatttcaatttgttataaattcctataatttatcctattagtagctaaaaa
aagatgaatgtgaatcgaatcctaagagaattaatgatacggcgaccaccgagatctaca
ctctttccctacacgacgctcttccgatctNNNNNNNNNNNNNNNNgcggccgcgagctcc
aattcgccctatagtgagtcgtattacaattcactgg
```

The "mutant RBD fragment" and "barcode fragment" were combined with NotI/SacI-digested parental plasmid backbone via HiFi Assembly. An example of the structure of the final assembled library (in the XBB.1.5 background) is available on GitHub: https://github.com/tstarrlab/SARS-CoV-2-RBD_DMS_Omicron-XBB-BQ/blob/main/data/186lib_pETcon_SARS2_XBB1-5.gb. Assembled library plasmids were electroporated into *E. coli* (NEB 10-beta, New England Biolabs C3020K), and plated at limiting dilutions on LB+ampicillin plates. For each library, duplicate plates corresponding to an estimated bottleneck of ~60,000 (XBB.1.5) and ~90,000 (BQ.1.1) cfu were scraped and plasmid purified. For positions that failed mutagenesis QC from Twist, in-house mutagenesis pools for each position were constructed via PCR with NNS mutagenic primers, HiFi assembled, plated to approximately ~400 cfu per position, and plasmid purified and pooled with the primary plasmid library. Plasmid libraries are available from Addgene (Addgene Cat # 1000000231 and 1000000232). Plasmid libraries were transformed into the AWY101 yeast strain [52] at 10-μg scale according to the protocol of Gietz and Schiestl [53], and aliquots of 9 OD*mL of yeast outgrowth were flash frozen and stored at -80˚C.

As described previously [6,8,18,51], we sequenced NotI-digested plasmid libraries on a Pac-Bio Sequel IIe to generate long sequence reads spanning the N16 barcode and mutant RBD coding sequence. The resulting circular consensus sequence (CCS) reads are available on the NCBI Sequence Read Archive (SRA), BioProject PRJNA770094, BioSample SAMN37185589. PacBio CCSs were processed using alignparse version 0.6.0 [54] to call N16 barcode sequence and RBD variant genotype and filter for high-quality sequences. Analysis of the PacBio sequencing indicates that all but 14 of the intended 4020 RBD mutations were sampled on ≥1 barcode in the BQ.1.1 libraries, and all but 8 were sampled on ≥1 barcode in the XBB.1.5 libraries with even coverage (S1B–S1E Fig). Complete computational pipelines and summary plots for PacBio data processing and library analysis are available on GitHub: https://github.com/tstarrlab/SARS-CoV-2-RBD_DMS_Omicron-XBB-BQ/blob/main/results/summary/process_ccs_BQ11.md and https://github.com/tstarrlab/SARS-CoV-2-RBD_DMS_Omicron-XBB-BQ/blob/main/results/summary/process_ccs_XBB15.md. Final barcode-variant lookup tables are available on GitHub: https://github.com/tstarrlab/SARS-CoV-2-RBD_DMS_Omicron-XBB-BQ/tree/main/results/variants

## Deep mutational scanning for ACE2-binding affinity

The effects of mutations on ACE2-binding affinity were determined via FACS-seq assays [6,18]. Titrations were performed in duplicate with pooled mutant libraries of Omicron BQ.1.1 and XBB.1.5 along with the BA.2 libraries constructed in [18]. Frozen yeast libraries were thawed, grown overnight at 30˚C in SD -Ura -Trp media (8 g/L Yeast Nitrogen Base, 2 g/L -Ura -Trp Synthetic Complete dropout powder and 2% w/v dextrose), and backdiluted to 0.67 OD600 in SG -Ura -Trp + 0.1%D (recipe as above but with 2% galactose and 0.1% dextrose in place of the 2% dextrose) to induce RBD expression, which proceeded for 22–24 hours at 19˚C with gentle mixing.

Induced cells were washed with PBS-BSA (BSA 0.2 mg/L), split into 16-OD*mL aliquots, and incubated with biotinylated monomeric human ACE2 protein (ACROBiosystems AC2-H82E8) across a concentration range from $10^{-6}$ to $10^{-13}$ M at 1-log intervals, plus a 0 M sample. Incubations equilibrated overnight at 19˚C with gentle mixing. Yeast were washed twice with ice-cold PBS-BSA and fluorescently labeled for 1 hr at 4˚C with 1:100 FITC-conjugated chicken anti-Myc (Immunology Consultants CMYC-45F) to detect yeast-displayed RBD protein and 1:200 PE-conjugated streptavidin (Thermo Fisher S866) to detect bound ACE2. Cells were washed and resuspended in 1x PBS for flow cytometry.

At each ACE2 sample concentration, single RBD⁺ cells were partitioned into bins of ACE2 binding (PE fluorescence) as shown in S2A Fig using a BD FACSAria II. A minimum of 10 million cells were collected at each sample concentration, and sorted into SD -Ura -Trp with pen-strep antibiotic and 1% BSA. Collected cells in each bin were grown overnight in 1 mL SD -Ura -Trp + pen-strep, and plasmid was isolated using a 96-well yeast miniprep kit (Zymo D2005) according to kit instructions, with the addition of an extended (>2 hr) Zymolyase treatment and a -80˚C freeze/thaw prior to cell lysis. N16 barcodes in each post-sort sample were PCR amplified as described in [8] and submitted for Illumina NextSeq P2 sequencing. Barcode reads are available on the NCBI SRA, BioProject PRJNA770094, BioSample SAMN37185929.

Demultiplexed Illumina barcode reads were matched to library barcodes in barcode-mutant lookup tables using dms_variants (version 1.4.3), yielding a table of counts of each barcode in each FACS bin, available at https://github.com/tstarrlab/SARS-CoV-2-RBD_DMS_Omicron-XBB-BQ/blob/main/results/counts/variant_counts.csv. Read counts in each FACS bin were downweighted by the ratio of total sequence reads from a bin to the number of cells that were sorted into that bin from the FACS log.

We estimated the level of ACE2 binding of each barcoded mutant at each ACE2 concentration based on its distribution of counts across FACS bins as the simple mean bin [8]. We determined the ACE2-binding constant $K_D$ for each barcoded mutant via nonlinear least-squares regression using the standard non-cooperative Hill equation relating the mean sort bin to the ACE2 labeling concentration and free parameters $a$ (titration response range) and $b$ (titration curve baseline):

$$\text{bin} = a \times [\text{ACE2}]/([\text{ACE2}] + K_D) + b$$

The measured mean bin value for a barcode at a given ACE2 concentration was excluded from curve fitting if fewer than 2 counts were observed across the four FACS bins or if counts exhibited bimodality (>40% of counts of a barcode were found in each of two non-consecutive bins). To avoid errant fits, we constrained the value $b$ to (1, 1.5), $a$ to (2, 3), and $K_D$ to ($10^{-15}$, $10^{-5}$). The fit for a barcoded variant was discarded if the average cell count across all sample concentrations was below 2, or if more than one sample concentration was missing. We also discarded curve fits where the normalized mean square residual (residuals normalized relative to the fit response parameter $a$) was >30 times the median value across all titration fits. Final binding constants were expressed as $-\log_{10}(K_D)$, where higher values indicate higher binding affinity. The complete computational pipeline for calculating and filtering per-barcode binding constants is available at https://github.com/tstarrlab/SARS-CoV-2-RBD_DMS_Omicron-XBB-BQ/blob/main/results/summary/compute_binding_Kd.md, and per-barcode affinity values are available at https://github.com/tstarrlab/SARS-CoV-2-RBD_DMS_Omicron-XBB-BQ/blob/main/results/binding_Kd/bc_binding.csv.

The affinity measurements of replicate barcodes representing an identical amino acid mutant were averaged within each experimental duplicate. The correlations in collapsed affinities in each duplicate experiment are shown in S2B Fig. The final measurement was determined as the average of duplicate measurements. The median mutant's final ACE2 affinity measurement collapsed across ~25 total replicate barcodes (S1E Fig). The final $-\log_{10}(K_D)$ for each mutant and number of replicate barcode collapsed into this final measurement for each RBD mutant are given in S1 Data and https://github.com/tstarrlab/SARS-CoV-2-RBD_DMS_Omicron-XBB-BQ/blob/main/results/final_variant_scores/final_variant_scores.csv, which includes data also from prior SARS-CoV-2 variant DMS datasets [6,18].

## RBD expression deep mutational scanning

Pooled libraries were grown and induced for RBD expression as described above. Induced cells were washed and labeled with 1:100 FITC-conjugated chicken anti-Myc to label for RBD expression via a C-terminal Myc tag, and washed in preparation for FACS. Single cells were partitioned into bins of RBD expression (FITC fluorescence) using a BD FACSAria II as shown in S3A Fig. A total of >13 million viable cells (estimated by plating dilutions of post-sort samples) were collected across bins for each library. Cells in each bin were grown out in SD -Ura -Trp + pen-strep, plasmid isolated, and N16 barcodes sequenced as described above. Barcode reads are available on the NCBI SRA, BioProject PRJNA770094, BioSample SAMN37185929.

Demultiplexed Illumina barcode reads were matched to library barcodes in barcode-mutant lookup tables using dms_variants (version 0.8.9), yielding a table of counts of each barcode in each FACS bin, available at https://github.com/tstarrlab/SARS-CoV-2-RBD_DMS_Omicron-XBB-BQ/blob/main/results/counts/variant_counts.csv. Read counts in each bin were downweighted using the post-sort colony counts instead of the FACS log counts as with ACE2 titrations above to account for unequal viability of cells in FITC fluorescence bins (i.e.,

many cells in bin 1 are non-expressing because they have lost the low-copy expression plasmid and do not grow out post-FACS in selective media).

We estimated the level of RBD expression (log-mean fluorescence intensity, logMFI) of each barcoded mutant based on its distribution of counts across FACS bins and the known log-transformed fluorescence boundaries of each sort bin using a maximum likelihood approach [8,55] implemented via the fitdistrplus package in R [56]. Expression measurements were discarded for barcodes for which fewer than 10 counts were observed across the four FACS bins. The full pipeline for computing per-barcode expression values is available at https://github.com/tstarrlab/SARS-CoV-2-RBD_DMS_Omicron-XBB-BQ/blob/main/results/summary/compute_expression_meanF.md, and per-barcode expression measurements are available at https://github.com/tstarrlab/SARS-CoV-2-RBD_DMS_Omicron-XBB-BQ/blob/main/results/expression_meanF/bc_expression.csv. Final mutant expression values were collapsed within and across replicates as described above, with correlation between experimental replicates shown in S3B Fig. Final mutant expression values and number of replicate barcode collapsed into this final measurement for each RBD mutant are available in S1 Data https://github.com/tstarrlab/SARS-CoV-2-RBD_DMS_Omicron-XBB-BQ/blob/main/results/final_variant_scores/final_variant_scores.csv.

## Quantification of epistasis

Epistatic shifts at each site between pairs of RBD variants were quantified exactly as described by [6]. Briefly, affinity phenotypes of each mutant at a site were transformed to a probability analog via a Boltzmann weighting, and the "epistatic shift" metric was calculated as the Jensen-Shannon divergence between the vectors of 21 amino acid probabilities (including the deletion character, when present). The Jensen-Shannon divergence ranges from 0 for two vectors of probabilities that are identical to 1 for two vectors that are completely dissimilar. To avoid noisier measurements artifactually inflating the epistatic shift metric, a given amino acid mutation was only included in the computation if it was sampled with a minimum of 3 replicate barcodes in each RBD background being compared. The calculation of epistatic shifts can be found at https://github.com/tstarrlab/SARS-CoV-2-RBD_DMS_Omicron-XBB-BQ/blob/main/results/summary/epistatic_shifts.md.

To visually depict epistatic shifts in mutational effects over SARS-CoV-2 evolution, we used multidimensional scaling to plot epistatic dissimilarities between each pair of RBDs for which we have human ACE2-binding deep mutational scanning measurements. We computed dissimilarities as the root-mean-squared epistatic shift across the 201 RBD sites (i.e., the dissimilarity between Wuhan-Hu-1 and Omicron BA.2 is computed as the root mean square of the y-axis metric illustrated in the top panel of Fig 3A). Root-mean-squared epistatic shifts minimize the influence of small-effect sites that are more likely to report on experimental noise or inconsequential epistatic shifts. We obtained qualitatively similar multidimensional scaling layouts if we computed dissimilarities as the simple arithmetic mean of epistatic shift metrics across RBD sites (where sites with smaller epistatic shifts have larger impacts) or if we computed L3- or L4-norm summed epistatic shifts (where sites with smaller epistatic shifts have further diminished impacts), which can be seen at https://github.com/tstarrlab/SARS-CoV-2-RBD_DMS_Omicron-XBB-BQ/blob/main/results/summary/epistatic_shifts.md#multidimensional-scaling-visualization-of-epistatic-drift.

## Natural variation in SARS-CoV-2 sequences

We downloaded the 16,238,458 spike sequences available from GISAID on November 11, 2023, and removed accessions from non-human origin. We discarded any spike sequence

shorter than 1260 or longer than 1276 amino acids and we removed an accession if it contained any ambiguous spike residues, leaving 9,811,651 final sequences. We aligned sequences via MAFFT and truncated spike sequences to RBD (N331-T531 in SARS-CoV-2 Wuhan-Hu-1 numbering), and further removed any accessions with >1 single-codon gap in the RBD sequence, leaving a final alignment of 9,811,285 RBD sequences. We then computed the number of times each individual mutation (including deletions) was present in the alignment relative to the Wuhan-Hu-1 reference sequence.

## Sites of strong antibody escape

Sites of strong antibody escape are determined from a large aggregation of deep mutational scanning antibody-escape datasets summarized in [30], available from https://github.com/jbloomlab/SARS2_RBD_Ab_escape_maps/ (note that this repository has since been replaced by a newer repository integrating additional antibody-escape deep mutational scanning datasets gathered in more updated Omicron backgrounds). We downloaded the aggregate dataset on June 1, 2023, and highlighted sites as "sites of strong escape" if their normalized site-wise escape score averaged across all mAbs was greater than 0.1.

## Supporting information

**S1 Fig. Deep mutational scanning library construction.** (**A**) Scheme for the generation of Omicron BQ.1.1 and XBB.1.5 mutant libraries. Site saturation mutagenesis oligonucleotide libraries were constructed by Twist Bioscience with constant flank sequences and cloned via three-fragment Gibson Assembly to create libraries of barcoded mutant variants. PacBio sequencing of the barcoded mutant library plasmid was used to create barcode:variant lookup tables, enabling subsequent Illumina sequencing of barcode fragments in experimental partitions to generate mutant phenotypes. (**B-E**) For pooled duplicate BQ.1.1 (top) and XBB.1.5 (bottom) libraries, we show (B) the per-variant rate of mutation types, (C) the distribution of number of amino acid mutants per barcoded variant, (D) the mutation rate of each type along each site in the RBD sequence, and (E) the distribution of total number of barcodes that were averaged for each amino acid mutant in the final ACE2-binding score.
(PNG)

**S2 Fig. Deep mutational scanning for ACE2-binding affinity.** (**A**) Representative FACS binning scheme for ACE2-binding titration assays. Bins of PE fluorescence (ACE2 binding) were drawn on cells pre-selected on FSC/SSC and FITC(RBD)/FSC plots to isolate single RBD-positive cells. At each ACE2 concentration, >10 million cells were collected in total across the four bins. Post-sort cells were sequenced to enable deconvolution of the titration of each barcoded library variant in parallel. (**B**) Correlation in mutant ACE2-binding affinities in experimental duplicates (independently barcoded and assayed mutant libraries). Red dashed line indicates the 1:1 linear line. (**C, D**) Relationship between amino acid mutant (C) and deletion (D) affinities in the Omicron BQ.1.1 versus XBB.1.5 backgrounds.
(PNG)

**S3 Fig. Deep mutational scanning measurements of mutational effects on folded RBD expression.** (**A**) Representative FACS scheme used for RBD expression deep mutational scans. Bins of FITC fluorescence (RBD expression) were drawn on cells pre-selected on FSC/SSC plots to isolate single cells. Cells were collected across the four bins and sequenced to identify the distribution of each library variant across bins and calculation of per-variant expression (mean fluorescence intensity, MFI). (**B**) Correlation in mutant RBD expression measurements in experimental duplicates (independently barcoded and assayed mutant libraries). Red

dashed line represents the 1:1 linear line. (**C**) Heatmaps illustrate the impacts of all mutations in the BQ.1.1 and XBB.1.5 RBDs on RBD expression.
(PNG)

**S4 Fig. Mutation-level epistatic shifts.** (**A**) Scatterplots of per-mutant affinities as in Fig 3B, illustrating nonspecific epistasis with impacts of mutations at buried core positions between XBB.1.5 and BA.2, with modestly affinity-decreasing mutations showing amplified effects in the BA.2 background. (**B**) Scatterplots of per-mutant affinities at sites 453, 455, 456 as shown in Fig 3B but for BQ.1.1 data. Orange letters indicate mutants that were assayed with fewer than three replicate barcodes in one of the two backgrounds; these mutants are excluded from the computation of the epistatic shift metric to reduce impacts of noise on the aggregate epistatic shift metric. (**C**) Representative binding curves from one of the three replicates of epistasis validation illustrated in Fig 3D. Yeast-displayed RBD mutants were expressed, incubated across a concentration series of monomeric human ACE2, and PE fluorescence (ACE2 binding) was determined via flow cytometry. Standard hill curves were fit to infer the midpoint EC50 binding constant.
(PNG)

**S1 Data. The effects of all single amino acid mutations in the Omicron BQ.1.1 and XBB.1.5 RBD on ACE2-binding affinity and RBD expression.** These data are also available at: https://github.com/tstarrlab/SARS-CoV-2-RBD_DMS_Omicron-XBB-BQ/blob/main/results/final_variant_scores/final_variant_scores.csv.
(CSV)

## Acknowledgments

We thank the High-Throughput Genomics Shared Resource at the University of Utah Huntsman Cancer Institute supported by NCI/NIH (P30CA042014), the Flow Cytometry Core Facility at the University of Utah Health Sciences Campus supported by NIH (S10OD026959 and 5P30CA042014-24), the University of Utah Center for High Performance Computing, supported by NIH (1S10OD021644-01A1), and the Fred Hutchinson Cancer Center Genomics core facility for experimental support. We thank Jesse Bloom and Bernadeta Dadonaite for helpful discussions. We thank all contributors of sequence data to GISAID.

## Author Contributions

**Conceptualization:** Tyler N. Starr.

**Formal analysis:** Ashley L. Taylor, Tyler N. Starr.

**Funding acquisition:** Tyler N. Starr.

**Investigation:** Ashley L. Taylor, Tyler N. Starr.

**Methodology:** Tyler N. Starr.

**Software:** Tyler N. Starr.

**Supervision:** Tyler N. Starr.

**Validation:** Ashley L. Taylor.

**Visualization:** Tyler N. Starr.

**Writing – original draft:** Tyler N. Starr.

**Writing – review & editing:** Ashley L. Taylor, Tyler N. Starr.

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
