## [Decision Letter · Decision Letter 0]

15 Nov 2023

Dear Assistant Professor Starr,

Thank you very much for submitting your manuscript "Deep mutational scans of XBB.1.5 and BQ.1.1 reveal ongoing epistatic drift during SARS-CoV-2 evolution" for consideration at PLOS Pathogens. As with all papers reviewed by the journal, your manuscript was reviewed by members of the editorial board and by several independent reviewers. The reviewers appreciated the attention to an important topic. Based on the reviews, we are likely to accept this manuscript for publication, providing that you modify the manuscript according to the review recommendations.

Sincerely,

Sergei L. Kosakovsky Pond, PhD

Academic Editor

PLOS Pathogens

Ronald Swanstrom

Section Editor

PLOS Pathogens

Kasturi Haldar

Editor-in-Chief

PLOS Pathogens

orcid.org/0000-0001-5065-158X

Michael Malim

Editor-in-Chief

PLOS Pathogens

orcid.org/0000-0002-7699-2064

Reviewer Comments (if any, and for reference):

Reviewer's Responses to Questions

**Part I - Summary**

Reviewer #1: The manuscript by Taylor and Starr uses their established deep mutational scanning technique to analyze the ongoing epistatic drift in the background of XBB.1.5 and BQ.1.1 lineages. Overall I think this is an important study that is well performed and well written. My comments are mostly designed to make the manuscript easier for the reader.

Reviewer #2: Taylor and Starr have performed deep mutational scans of XBB.1.5 RBD and BQ.1.1 RBD. The mutant libraries included single amino acid deletion, which is innovative and generated informative results. Epistatic shifts in both XBB.1.5 RBD and BQ.1.1 RBD were identified. The authors have further shown that the amino acid difference between XBB.1.5 and BQ.1.1 at position 493 modulated the effects of Y453W, L455W, and F456L on ACE2-binding. The analysis of epistatic drift at the end showed the importance of positions 498 and 501 in determining the epistatic landscape. Overall, this study is very nicely done and provides important insights into the SARS-CoV-2 evolution. I only have minor comments.

**Part II – Major Issues: Key Experiments Required for Acceptance**

Reviewer #1: Figure 1a. I appreciate why the authors did not want to list all of the mutations for each lineage but the figure is going to mislead a lot of readers because they aren’t going to see the note. The authors need to find a way to visually demonstrate that the BA.2 mutations are still in the BQ/XBB. Perhaps extending the lines from the mutations in BA.2. downward.

Figure 1b. This is the meat of the manuscript, but it would be impossible to read in printed form (some people still print papers for reading). Recommend separating into two and making it a full-page figure.

Figure 2F is a problem. Without some more careful curating this is subfigure is only an illustration of where sequence dropouts are most likely to occur. I did a few spot checks of the sequences and noticed a few patterns. 1) the deletions were mostly not isolated but part of bigger deletions which are almost certainly sequencing artifacts. For instance, over 80% of the sequences with 482DEL (the most prevalent on their chart) also had 483DEL. 2) the sequences with the deletions mostly came from a handful of sequencing facilities. For instance, Canary islands make up less than 0.1% of all sequences in GISAID, but managed to generate 5-10% of the sequences with deletions (from the ones I checked). One partial solution to this would be to look for the sequences that have discrete deletions (I.E., search for sequences lacking one AA but not lacking the two flanking AAs). This isn’t perfect, but would get rid of most of the crap and it would be more of an ‘apples to apples’ comparison with the manuscript deletions. Alternatively they could restrict it to lineages with the same deletion more than once in the same background, but that would be a lot harder to compute. The subfigure is unacceptable in its current form.

Reviewer #2: None

**Part III – Minor Issues: Editorial and Data Presentation Modifications**

Reviewer #1: Line 61-62. “in part to its ability to balance strength of ACE2 binding with immune escape via the beneficial F486P mutation”. Could the authors elaborate on what they mean by this. In their predictor F486P is not predicted to enhance ACE2 binding of BA.2 (the parent of XBB), but it is predicted to enhance binding of BQ.1.1, which never acquired it.

Line 139, reference Fig 1B. With the microprint I was initially waiting for the figure with the deletion data not realizing it was one of the lines in 1B.

Line 171. Period missing.

Line 200. Confusing way of stating.

Line 315. What’s are epistatic shits?

Reviewer #2: 1. Lines 34-36: “… the spike receptor-binding domain (RBD) has shown a particularly high rate of sequence evolution due to its role in binding ACE2 receptor to enable cell entry [2,3] …” It is unclear to me how the evolution of SARS-CoV-2 RBD is driven by its role in ACE2 binding.

2. Lines 202-203: “Instead, these sites show greater sensitization to deleterious mutational effects in the BA.2 background (Supplemental Figure 4A).” It is a bit unclear to me how this conclusion can be drawn based on Supplemental Figure 4A. Additional explanation will be helpful. Maybe adding a gray dashed line indicating the additive (non-epistatic) expectation, just like Figure 3B, will also help.

3. Lines 268-269: “… as well as the divergent human-ACE2-utilizing SARS-CoV-1 RBD …” Reference needed.

4. Lines 303-304: “… early fixation of Q498 substitutions would have instead driven SARS-CoV-2 closer to SARS-CoV-1 in the epistatic landscape.” Do the authors think that any aromatic amino acid substitutions (F, Y, W) at position 498 would similarly drive SARS-CoV-2 closer to SARS-CoV-1?

PLOS authors have the option to publish the peer review history of their article (what does this mean?). If published, this will include your full peer review and any attached files.

Reviewer #1: No

Reviewer #2: No

Figure Files:

Data Requirements:

Reproducibility:

References:

---

## [Editor Report · Decision Letter 1]

14 Dec 2023

Dear Assistant Professor Starr,

We are pleased to inform you that your manuscript 'Deep mutational scans of XBB.1.5 and BQ.1.1 reveal ongoing epistatic drift during SARS-CoV-2 evolution' has been provisionally accepted for publication in PLOS Pathogens.

Best regards,

Sergei L. Kosakovsky Pond, PhD

Academic Editor

PLOS Pathogens

Ronald Swanstrom

Section Editor

PLOS Pathogens

Kasturi Haldar

Editor-in-Chief

PLOS Pathogens

orcid.org/0000-0001-5065-158X

Michael Malim

Editor-in-Chief

PLOS Pathogens

orcid.org/0000-0002-7699-2064
---

## [Editor Report · Acceptance letter]

22 Dec 2023

Dear Assistant Professor Starr,

We are delighted to inform you that your manuscript, "Deep mutational scans of XBB.1.5 and BQ.1.1 reveal ongoing epistatic drift during SARS-CoV-2 evolution," has been formally accepted for publication in PLOS Pathogens.

Best regards,

Michael Malim

Editor-in-Chief

PLOS Pathogens

orcid.org/0000-0002-7699-2064